# Effects of Starch Level and a Mixture of Sunflower and Fish Oils on Nutrient Intake and Digestibility, Rumen Fermentation, and Ruminal Methane Emissions in Dairy Cows

**DOI:** 10.3390/ani11051310

**Published:** 2021-05-02

**Authors:** Babak Darabighane, Ilma Tapio, Laura Ventto, Piia Kairenius, Tomasz Stefański, Heidi Leskinen, Kevin J. Shingfield, Johanna Vilkki, Ali-Reza Bayat

**Affiliations:** 1Animal Nutrition, Production Systems, Natural Resources Institute Finland (Luke), FI-31600 Jokioinen, Finland; ext.babak.darabighane@luke.fi (B.D.); laura.ventto@gmail.com (L.V.); piia.kairenius@luke.fi (P.K.); tomasz.stefanski@luke.fi (T.S.); heidi.leskinen@luke.fi (H.L.); kevin.shingfield@luke.fi (K.J.S.); 2Genomics and Breeding, Production Systems, Natural Resources Institute Finland (Luke), FI-31600 Jokioinen, Finland; ilma.tapio@luke.fi; 3Research and Customer Relationships, Service Groups, Natural Resources Institute Finland (Luke), FI-31600 Jokioinen, Finland; johanna.vilkki@luke.fi

**Keywords:** starch, lipid, methane, microbial diversity

## Abstract

**Simple Summary:**

Methane produced by ruminants contributes to increased greenhouse gas effect. There are various nutritional strategies to reduce methane emission, such as supplementing fat or changing starch levels in the diet. Understanding the interactions of these strategies on methane emission, as well as performance, digestibility, and rumen fermentation is important. The present study aimed to assess the effects of starch level with or without a mixture of sunflower and fish oils on nutrient intake and digestibility, milk yield and composition, rumen fermentation, ruminal CH_4_ emissions and microbial ecology in dairy cows. Oil mixture rich in polyunsaturated fatty acids supplemented to low- or high-starch diets reduced dry matter intake and increased energy digestibility of lactating cows. High starch level improved nutrient digestibility and tended to reduce ruminal acetate:propionate ratio but did not affect rumen pH, molar propionate ratio, or ruminal CH_4_ emissions. Oil decreased absolute ruminal CH_4_ emission or tended to decrease CH_4_ per energy corrected milk.

**Abstract:**

Four multiparous dairy cows were used in a 4 × 4 Latin square to examine how starch level and oil mixture impact dry matter (DM) intake and digestibility, milk yield and composition, rumen fermentation, ruminal methane (CH_4_) emissions, and microbial diversity. Experimental treatments comprised high (HS) or low (LS) levels of starch containing 0 or 30 g of a mixture of sunflower and fish oils (2:1 *w*/*w*) per kg diet DM (LSO and HSO, respectively). Intake of DM did not differ between cows fed LS and HS diets while oil supplementation reduced DM intake. Dietary treatments did not affect milk and energy corrected milk yields. There was a tendency to have a lower milk fat concentration due to HSO compared with other treatments. Both high starch level and oil supplementation increased digestibility of gross energy. Cows receiving HS diets had higher levels of total rumen VFA while acetate was lower than LS without any differences in rumen pH, or ruminal CH_4_ emissions. Although dietary oil supplementation had no impact on rumen fermentation, decreased CH_4_ emissions (g/day and g/kg milk) were observed with a concomitant increase in *Anoplodinium-Diplodinium* sp. and *Epidinium* sp. but a decrease in *Christensenellaceae*, *Ruminococcus* sp., *Methanobrevibacter ruminantium* and *Mbb. gottschalkii* clades.

## 1. Introduction

The growing human population is boosting the demand for milk and meat as sources of animal protein, resulting in several challenges for ruminant production systems, including the need to reduce their contribution to greenhouse gas emissions. This calls for special attention to solutions for reducing methane (CH_4_) emissions from ruminants without negative effects on productivity. Additionally, it has been shown that 5–7% of gross energy (GE) intake is lost through CH_4_ production from dairy cows [1]. A number of strategies, including management, dietary approaches, and genetics have been proposed for CH_4_ mitigation [2,3,4]. In fact, chemical composition of the feed [5] and changing the starch content of the concentrate has been proposed as a CH_4_ reducing strategy [2,4]. This effect was attributed to an increase in the number of amylolytic bacteria and a drop in the number of methanogens and fibrolytic bacteria, changing ruminal volatile fatty acids (VFA) in favor of propionate production [3,4] and creating an alternative hydrogen sink to methanogenesis [6].

On the other hand, unsaturated fat supplementation is another feeding strategy which not only reduces enteric CH_4_ production [1,2] but can improve milk monounsaturated (MUFA) and polyunsaturated fatty acid (PUFA) composition [7] with potential benefits to human health [8]. Unlike starch’s mode of action, lipid sources are not fermented in the rumen; rather, they lower fermented organic matter (OM), leading to a drop in CH_4_ production. Furthermore, it has been shown that medium-chain fatty acids (C14–C17) also affect the number of methanogens while unsaturated fatty acids (linoleic and α-linolenic acids) shift rumen fermentation towards production of propionate and, therefore, reduce CH_4_ production through toxic effects on cellulolytic bacteria and protozoa [4].

Thus, our hypothesis was that higher starch level and oil supplementation would have additive effects on reducing ruminal CH_4_ production in dairy cows. Therefore, the present study aimed to assess the effects of starch level with or without a mixture of unsaturated fatty acids (sunflower and fish oils) on nutrient intake and digestibility, milk yield and composition, rumen fermentation, ruminal CH_4_ emissions, and microbial ecology in dairy cows.

## 2. Materials and Methods

All experimental procedures were approved by the National Ethics Committee (ESAVI/4342/04.10.03/2011, Turku, Finland) in accordance with the guidelines established by the European Community Council Directive 86/609/EEC [9].

### 2.1. Animals, Experimental Design and Diets

A 4 × 4 Latin square with 2 × 2 factorial arrangement of treatments was applied to four multiparous Nordic Red cows in mid-lactation (76 ± 10.4 days in milk; mean ± SD) producing 35.2 ± 2.10 kg milk/d. The cows were fitted with rumen cannula (#1C, i.d. 100 mm; Bar Diamond Inc., Parma, ID, USA) and each experimental period consisted of 14 days diet adaptation, five days as sampling period, and a 16-day washout to avoid carry-over effects to the next period. The cows were randomly allocated to the diets. Diets, formulated to be isonitrogenous, were used based on grass silage (forage to concentrate ratio 55:45 on a dry matter (DM) basis) consisting of low starch (LS) or high starch (HS) levels (16.1 and 202 g/kg DM) with 0 or 30 g of unsaturated fatty acid mixture (sunflower oil-fish oil mixture; 2:1 *w*/*w*)/kg diet DM (LSO and HSO, respectively) (Table 1). Sugar beet pulp and barley feed in LS diets were replaced with rolled barley and ground wheat to provide different levels of starch, and urea was added to make the diets as iso-nitrogenous as possible.

The oils were stored in +4 °C until incorporated into the low-or high-starch TMR to avoid oxidation of unsaturated fatty acids and the oil replaced concentrate ingredients. The oil mixture and starch levels were selected based on our previous experiences [10] where satisfactory induction of milk fat depression was realized. One of the main objectives of this work was to study milk fat depression phenomenon (not reported in this paper). The grass silage was prepared from timothy and meadow fescue (54:46) grown at Jokioinen (60°49′ N, 23°28′ E), and ensiled with a formic acid-based ensiling additive (AIV2 plus, 5 L/t; AIV Valio Ltd., Helsinki, Finland) to allow for a restricted fermentation. In order to avoid selection of dietary components and to maintain the target forage to concentrate ratio, the diets were prepared as TMR. Experimental diets were formulated to meet or exceed metabolizable energy and protein requirements of lactating cows producing 30 kg milk/d [11] offered ad libitum to result in 5–10% refusals, and fed in four equal amounts at 06:00 h, 09:00 h, 16:30 h, and 19:30 h. Cows were kept in individual tie stalls, had free access to water and salt blocks, and were milked at 07:00 h and 16:45 h.

### 2.2. Feed Intake, Milk Yield and Chemical Analysis

Daily feed intake was measured by subtracting the refusals from the offered feed throughout the study but intakes during d 14–17 of each experimental period were used for statistical analysis. Representative samples of silage and concentrate ingredients during the sample collection period were used for chemical analysis. The samples were pooled within each period before chemical analysis using the standard methods described by Shingfield et al. [12]. In addition, the method proposed by Huida et al. [13] was used to correct silage DM content for the loss of volatiles. Indigestible neutral detergent fiber (iNDF) of silage and concentrates was determined by 12 d of ruminal incubation using nylon bags (60 × 120 mm, pore size 0.017 mm; Swiss Silk Bolting Cloth Mfg. Co. Ltd., Zurich, Switzerland) followed by neutral detergent fiber (NDF) analysis excluding ash. Chemical analysis of silage, concentrates, and oils plus their proportion in each diet were used to calculate chemical composition of each experimental TMR. Bomb calorimetry (1108 Oxygen bomb, Parr Instrument Co., Moline, IL, USA) was conducted to determine the GE of silage, concentrates, oils, and excreta. Milk samples were collected over 10 consecutive milking during d 15–19 of each experimental period, treated with preservative (Bronopol, Valio Ltd., Helsinki, Finland). Milk fat, crude protein (CP), and lactose were predicted using infrared analysis (MilkoScan 133B, Foss Electric, Hillerød, Denmark). Milk composition was calculated based on the weighted average of morning and afternoon milk yields.

### 2.3. Rumen Fermentation

On d 18 of each period and at 1.5 h intervals from 06:00 until 16:30 h, a suction pump with a Büchner flask was used to collect samples of ruminal fluid (150 mL; *n* = 8) through the rumen cannula. Then, pH was measured using a portable pH meter (pH 110, VWR International). Rumen liquid was filtered through two layers of cheesecloth and 5.0 mL ruminal fluid was preserved with 0.5 mL of saturated HgCl_2_ and 2.0 mL of 1 M NaOH to determine VFA. Furthermore, ammonia-N concentration was analyzed by collecting additional ruminal fluid (15.0 mL) preserved with 0.3 mL of 50% (*vol*/*vol*) sulfuric acid. The ruminal samples were stored at −20 °C until the time of analysis. Analysis of VFA and ammonia-N were performed as described by Shingfield et al. [12].

### 2.4. Apparent Total-Tract Digestibility

Feces were collected over a 72-h interval starting at 18:00 h on d 14 of each experimental period and then used to determine total tract apparent digestibility coefficients. A light harness and flexible tubing attached to the vulva was used to separate urine and feces. Representative fecal samples were collected daily and composited, dried in an oven (55 °C, 48 h). The chemical composition of fecal samples was determined using the same methods for the feed samples as described earlier.

### 2.5. Ruminal Gas Production

Ruminal CH_4_ and carbon dioxide (CO_2_) emissions were recorded over 6 days period (d 11–17 of each period) using sulfur hexafluoride (SF_6_) as a tracer marker as described by Bayat et al. [14] and validated against respiration chambers by Bayat et al. [15]. Briefly, gases in the rumen headspace were drawn continuously (1.7 mL/min) over every 24-h period into evacuated 5.5 L air-tight canisters using a capillary tubing (PEEK 1.6 mm × 0.13 mm i.d., VICI Valcro Instruments Co, Houston, TX, USA). Tubes used to collect the ruminal gas were anchored securely to the neck of the rumen cannula allowing gas collection at approximately 5 cm above the rumen mat. No correction was made for background SF_6_, CH4 and CO_2_ concentrations because cows were housed in a well-ventilated facility (72 m^3^/min) and fitted with custom-made sponges placed between the outer edge of the cannula flange and the abdominal wall to minimize the exchange of surrounding air with ruminal contents. Gas chromatography (Agilent 6890N, Agilent Technologies, Santa Clara, CA, USA) proposed by Regina and Alakukku [16] was applied to sub-samples of ruminal gases to analyze them in triplicates for CH_4_, CO_2_, and SF_6_ concentrations. Actual release rate of SF_6_ (1.16 ± 0.19 mg/d) in the rumen during the experiment as well as the concentrations of CH_4_, CO_2_, and SF_6_ measured by GC were used to calculate daily ruminal CH_4_ and CO_2_ emissions as following:CH_4_ (L/d) = SF_6_ (L/d) × [CH_4_]/[SF_6_]
CO_2_ (L/d) = SF_6_ (L/d) × [CO_2_]/[SF_6_]

### 2.6. Microbial Analysis

Samples of ruminal digesta (2 kg) were collected from four sites (anterior dorsal, anterior ventral, posterior dorsal, and posterior ventral rumen sacs) during each period on d 15 at 15:00 h and d 17 at 09:00 h. Samples were mixed and 50 g subsample was placed into a plastic bag and snap-frozen in liquid nitrogen. Samples were stored at −80 °C until DNA extraction. Total DNA was extracted from 250 mg of ruminal digesta following Yu and Morrison [17] protocol. Rumen bacterial, archaeal and ciliate protozoan community composition was determined using 16S and 18S rRNA gene sequencing. Primers used for amplicon library preparation, sequencing conditions and sequencing data quality control were performed as described in Tapio et al. [18]. Sequencing data was further processed using Qiime v 1.9.1 [19]. Briefly, quality filtered sequences were clustered into operational taxonomic units (OTU) at 97% similarity using UCLUST [20]. Chimeric reads were filtered out using ChimeraSlayer [21]. Bacterial and archaeal OTUs taxonomy was assigned using the Greengenes 13_8 [22] and RIM-DB [23] databases, respectively. Ciliate protozoa OTUs were assigned using ciliate protozoa database [24]. Singleton OTUs were removed and the data from each sample were rarefied to the similar sequencing depth prior to further analyses.

### 2.7. Calculations

The difference between nutrient intake and fecal outputs was used to calculate total tract digestibility coefficients. Energy losses as CH_4_ were calculated using the factor 55.24 kJ/g [25]. Potentially digestible NDF (pdNDF) was calculated as the difference between NDF and iNDF. Energy corrected milk (ECM) yield was calculated according to Sjaunja et al. [26]. Methane (or CO_2_) emissions as proportions of organic matter intake (OMI), organic matter digestibility (OMD), and milk and ECM yields, were calculated by dividing daily CH_4_ emissions (g/d) by OMI, OMD, and milk and ECM yields, respectively. Methane emissions as percentage of GE intake (GEI) was calculated as CH_4_ energy (MJ/d) by GEI (MJ/d).

### 2.8. Statistical Analysis

Experimental data was analyzed by ANOVA for a 4 × 4 Latin square with a 2 × 2 factorial arrangement of treatments through the mixed procedure of SAS (version 9.2, SAS Inst. Inc., Cary, NC, USA) with a model that included fixed effects of period, starch level, oil level, and starch by oil interaction, and the random effect of cows assuming an autoregressive covariance structure fitted on the basis of Akaike information and Schwarz Bayesian model-fit criteria. The averages of data for cow within period were calculated before statistical analysis. The values reported are least square means ± SEM. The significance level *p* ≤ 0.05 was used to determine significant effects of starch, oil, and their interaction. In addition, probabilities at 0.05 < *p* < 0.10 were considered as a trend.

For microbial community analysis, taxa with less than 0.01% relative abundance across all samples were filtered out before further analyses. Data was normalized by cumulative-sum scaling and log_2_ transformation to account for the non-normal distribution of taxonomic count data, as implemented in Calypso [27]. Microbial community alpha diversity was estimated using Shannon, Simpson diversity indices, richness and evenness estimates. Redundancy analysis (RDA) and analysis of similarity (ANOSIM), calculated based on Bray-Curtis dissimilarities, were used to identify if diet, oil or starch can be explanatory factors for the rumen microbial community composition. Permutation multivariate analysis of dispersion (Permdisp2) was used to tests whether the dispersion between the groups is significant. Analysis of variance (ANOVA) followed by subsequent pairwise comparisons with Tukey test was performed to look at diet, oil or starch effect on individual taxa. Spearman correlation was used to explore associations between rumen fermentation, methane production phenotype data and individual microbial taxa. Comparisons were counted as significant with *p* < 0.05. *p* values were further adjusted for the false discovery rate (FDR).

## 3. Results

### 3.1. Dry Matter and Nutrient Intake and Milk Yield

Intakes of DM and GE did not differ (*p* ≥ 0.18) between cows fed LS and HS diets while oil supplementation reduced (*p* < 0.01) DM and GE intakes (Table 2). In comparison to LS diet, HS diet tended to increase (*p* = 0.087) OM intake while oil supplementation reduced (*p* < 0.01) OM intake. Starch intake was much greater for HS compared with LS diets as planned by design, but dietary oil inclusion reduced starch intake more in HS diet (*p* < 0.01 for the interaction of starch and oil). Intakes of NDF and water-soluble carbohydrates (WSC) were greater (*p* < 0.01) in cows fed LS than those fed HS diets. Intake of CP in cows fed HS diet was slightly greater (*p* < 0.05) and again oil supplementation led to a lower (*p* < 0.01) CP intake. As expected, intakes of saturated fatty acids (SFA), MUFA, and PUFA were greater (*p* < 0.01) with oil supplementation of both low- and high-starch diets. Milk and ECM yields were not influenced (*p* ≥ 0.11) by dietary treatments (Table 3). However, ECM yield was noticeably (2.7 kg/d) yet numerically lower for HSO diet than other treatments. There was a tendency (*p* = 0.07 for interaction of starch level and oil supplementation) to cause lower milk fat concentration due to HSO compared with other treatments and oil supplementation reduced (*p* < 0.01) milk protein concentration. Inclusion of oil in the diet tended to reduce (*p* = 0.087) milk fat yield and reduced (*p* < 0.05) milk protein yield. Milk production efficiency expressed as ECM/DMI was not affected (*p* ≥ 0.28) by dietary treatments.

### 3.2. Apparent Total-Tract Digestibility

The high starch level, but not oil supplementation, increased (*p* < 0.01) apparent total tract digestibility of DM, OM and starch while decreased (*p* < 0.01) NDF and pdNDF digestibility (Table 4). Both high starch level and oil supplementation increased (*p* ≤ 0.02) digestibility of GE and tended to increase (*p* = 0.058) CP digestibility. There was no interaction between starch level and oil supplementation for any of digestibility measurements.

### 3.3. Rumen Fermentation

The experimental treatments had no impact (*p* > 0.05) on rumen pH while HS diet tended to increase (*p* = 0.056) total VFA concentration compared with LS diets (Table 5). However, no significant change (*p* = 0.21) was observed in total VFA as a result of oil supplementation. Compared with the cows receiving LS diet, cows fed HS diets had lower (*p* < 0.01) molar proportion of acetate and greater (*p* < 0.05) molar proportions of butyrate, isobutyrate, valerate, isovalerate, and caproate. The experimental treatments did not influence (*p* ≥ 0.18) molar proportion for propionate. Acetate to propionate ratio tended to be lower (*p* = 0.07) for HS compared with LS diets. Ruminal ammonia-N was greater for HS compared with LS diets (*p* < 0.01), and dietary oil inclusion increased ammonia more in HS diet (*p* < 0.05 for the interaction of starch and oil).

### 3.4. Ruminal CH_4_ and CO_2_ Emission

Inclusion of the oil mixture in LS and HS diets reduced (*p* = 0.05) daily ruminal CH_4_ emissions (Table 6). Cows receiving oil supplements had lower CH_4_ emission intensity calculated as g/kg milk (*p* < 0.05) and g/kg ECM (*p* = 0.067) than their control counterparts. No difference (*p* ≥ 0.15) was found between the treatments in terms of CH_4_ emissions calculated as proportion of GE intake or g/kg OM digested. The experimental treatments were not different (*p* ≥ 0.16) in terms of daily ruminal CO_2_ emission and g/kg OM digested.

Ruminal CO_2_ emissions expressed as g/kg milk or ECM was greater for LS compared with other diets (*p* = 0.086 and 0.036 for the interaction, respectively).

### 3.5. Rumen Microbial Ecology

Sequencing yielded 5222–7362 good quality sequences per sample for bacteria, 1730–18,249 for archaea and 3339–12,123 for ciliate protozoa. Rumen bacterial community was represented by 18 phyla. Bacteroidetes (51–60%), Firmicutes (20–25%), Proteobacteria (1–12%), and Spirochaetes (0.9–5%) were the dominating phyla. Remaining phyla were detected at the abundance below 1%. Among bacterial genera, *Prevotella* was predominant in all dietary groups (40–50%) with other more abundant genera being unclassified *Succinivibrionaceae* (0–11%), unclassified *Clostridiales* (5–6%), *Treponema* (1–5%), unclassified *Ruminococcaceae* (3%), unclassified *Lachnospiraceae* (2–4%), *Succiniclasticum* (2%), *Ruminococcus*, *Fibrobacter*, CF231 and *Butyrivibrio* (altogether 1–2%). The remaining genera were detected at an abundance below 1%.

The Archaea community was dominated by *Methanobrevibacter gottschalkii* (50–63%) and *Methanobrevibacter ruminantium* (12–24%) in all the groups. Other more dominant archaea were *Methanosphaera* sp. ISO3F5 (6–24%), *Methanimicrococcus blatticola* (1–15%) and *Methanobacterium alkaliphilum* (1–3%). Archaea groups belonging to the Methanomassiliicoccaceae (Mmc) family were observed at the abundance below 1%.

Ciliate protozoa community was dominated by *Entodinium* (35–56%) in all dietary groups. Other predominant ciliate genera detected at the abundance above 5% in at least one of the diets were: *Polyplastron*, *Ostracodinium*, *Metadinium*, *Isotricha*, *Eudiplodinium-Eremoplastron*, *Epidinium*, *Charonina*, and *Anoplodinium-Diplodinium*.

Dietary treatments had little effect on microbial alpha diversity estimates. Only bacterial richness was significantly (*p* = 0.01) reduced in high starch diets and archaeal richness tended to be numerically lower (*p* = 0.06) in dietary treatments with oil additive. No significant diet, oil or starch effect was observed on richness of ciliate protozoa (Table 7).

Beta diversity analysis was performed using RDA and ANOSIM, and identified significant clustering of bacteria with respect to diet (*p* = 0.006) and starch (*p* = 0.006). Amount of starch in the diet was also a significant explanatory factor (*p* = 0.05) for the clustering of ciliate protozoa. No distinct clustering of archaea was identified due to diet, oil or starch. Beta dispersion was significantly different between high and low starch bacterial communities (*p* = 0.009).

### 3.6. Taxa Affected by Diet, Oil and Starch

Diets with oil additive significantly (*p* < 0.05) increased abundance of *Anaerovibrio* sp., (Spirochaetes) PL-11B10 and ciliate protozoa *Eudiplodinium-Eremoplastron* (AB536716). Oil caused a significant reduction in (Bacteroidetes) RF16, (Proteobacteria) GMD14H09, *Anaeroplasma* sp., *Prevotellaceae*, (TM7) F16, *Bacteroidales*, and archaea belonging to Mmc. Group 8 sp. WGK1. However, only increase in *Anaerovibrio* sp. remained significant after correction for multiple testing using false discovery rate (FRD = 0.035) (Figure 1, Appendix A).

Diets with high starch content had significantly higher relative abundance of bacteria (Bacteroidetes) S24-7, *Succinivibrionaceae* spp., *Ruminobacter* sp., *Selenomonas* sp., *Moryella* sp., *Ruminococcus flavefaciens*, and ciliate protozoa *Isotricha* sp. and *Entodinium* sp. Contrary, diets with low starch content were significantly enriched in *Treponema* sp., (Bacteroidetes) F16, SR1, *Lachnospira* sp., *Clostridium* sp., *Acholeplasmatales*, *Desulfovibrio* sp., (Tenericutes) RF39, (Actinobacteria) OPB41, *Lactobacillus* sp. and ciliates affiliated with *Eudiplodinium*-*Eremoplastron* (OTU12), *Charonina ventriculi*, *Ostracodinium* sp. and *Dasytricha* sp. After multiple testing correction, only (Bacteroidetes) S24-7, *Succinivibrionaceae* sp., *Isotricha* sp. and *Eudiplodinium*-*Eremoplastron* (OTU12) retained significant differences (Figure 1, Appendix A).

Diet composition had significant (*p* < 0.05) effect on 27 bacterial, one archaeal and seven ciliate protozoan taxa, but after FDR correction, only seven bacterial and two ciliate protozoan taxa remained significant (Figure 1). In pairwise comparisons, *Anaerovibrio* sp. was significantly more abundant in diets containing oil, in particular HSO diet when compared with HS, LS or LSO diets. *Eudiplodinium*-*Eremoplastron* (OTU12) was significantly more abundant in LS compared with HS or HSO diets, while *Isotricha* sp. was significantly more abundant in HS compared with LS or LSO diets. Bacteria (Bacteroidetes) F16, (Proteobacteria) GMD14H09, (Bacteroidetes) RF16 and SR1 were detected at the lowest abundance in HSO diet and showed significant differences between HSO and all other diets. Bacteria (Bacteroidetes) S24-7 had highest abundance in HSO diet and showed significant differences in pairwise comparisons with LS and LSO diets. Among archaea Mmc. Group 8 sp. WGK1 was more abundant (*p* = 0.03, FDR = 0.4) in HS diet compared with HSO or LSO diets (Figure 1, Appendix A).

### 3.7. Microbiota Association with Rumen Fermentation and Methane Production Traits

To look at the associations between rumen microbiome and rumen fermentation as well as methane production traits, we applied Spearman correlations on the microbial taxa present in all samples (n = 16 observations). From 50 bacterial, two archaeal and 25 ciliate protozoan significant associations (*p* < 0.05), eight bacterial and 10 ciliate protozoan correlations remained significant after correction for multiple testing using false discovery rate (FDR < 0.1).

An increase in acetate was negatively correlated with (Bacteroidetes) S24-7 (FDR = 0.005), but positively with *Clostridium* sp. (FDR = 0.035), ciliates *Dasytricha* sp. (FDR = 0.053), and *Charonina ventriculi* (FDR = 0.057) (Figure 2). Increase in ammonia-N was positively associated with (Bacteroidetes) S24-7 (FDR = 0.011) and *Moryella* sp. (FDR = 0.049) but negatively associated with *Treponema* sp. (FDR = 0.044), (Bacteroidetes) RF16 (FDR = 0.049), *Clostridium* sp. (FDR = 0.049), and *Charonina ventriculi* (FDR = 0.07). *Charonina ventriculi* was negatively correlated with butyrate (FDR = 0.043) while *Epidinium* sp. was negatively correlated with isobutyrate (FDR = 0.062) and isovalerate (FDR = 0.016). On the contrary, *Entodinium* sp and *Entodinium caudatum* were positively correlated with isovalerate (FDR = 0.089 and 0.016, respectively). *Anoplodinium*-*Diplodinium* sp. and *Epidinium* sp. were negatively correlated with daily CH_4_ emissions (FDR = 0.064), while *Christensenellaceae* and *Ruminococcus* sp. correlation was positive. *Ruminococcus flavefaciens* was positively correlated with valerate (FDR = 0.063). *Methanobrevibacter ruminantium* and *Mbb*. *gottschalkii* clades were both positively correlated with CH_4_ intensity (g/kg milk) (*p* < 0.03, FDR = 0.12) (Figure 2).

## 4. Discussion

### 4.1. Dry Matter Intake, Milk Yield, and Nutrient Digestibility

The results of this experiment showed that diets with oils rich in PUFA in a moderate amount reduced DM intake, which in turn resulted in lower intakes of other nutrients (OM, CP, NDF, WSC, and starch). Similar observations have been reported previously in experiments with PUFA-rich plant oils [28,29], fish oil [30,31], or a mixture of fish oil and plant seed oils [32,33,34]. In this experiment, dietary oil supplementation reduced DM intake by 9.7% on average. In line with these results, Shingfield et al. [33] reported approximately 20.5% reduction in DM intake as a result of adding a mixture of fish and sunflower oils to corn silage-based diet (45 g oil/kg DM and 63 g ether extract (EE)/kg DM) compared with a non-supplemented diet (EE content of 33.5 g/kg DM). However, another study reported no difference in terms of DM intake of cows between the control diet based on alfalfa and corn silage (27.8 g EE/kg DM) and the same diet supplemented with mixture of fish or canola oils (20 g oil/Kg DM and 46.7 g EE/kg DM, respectively; [31]). It has been well shown that the effect of oil supplementation of a diet on DM intake can be a function of combination of including oil content and diet composition, source of oil, and type of basal diet [29,35,36].

The lack of responses on milk yield due to dietary starch level and unsaturated oil mixture were expected as the diets were designed based on our previous experiences [10] to cause no effect on milk yield but lower fat and ECM yields due to the combination of high starch level and oil (i.e., HSO diet) known as milk fat depression effect. However, ECM yield was not influenced by dietary treatments despite being noticeably yet numerically lower for HSO diet (2.7 kg/d) compared with the average of other treatments. The reason for numerically lower ECM was the lower milk fat concentration (11% lower compared with LS diet).

The lack of oil supplementation effect on nutrient digestibility, with exception of CP and GE digestibility is consistent with some previous findings [14,37,38].

While in our experiment high-starch diet did not affect DM intake, an increase was observed for CP, OM, and starch intakes with drops in NDF and WSC intakes, which are due to differences in nutrient contents of the diets. The results of our study are consistent with the findings of Pirondini et al. [39] and Philippeau et al. [40]. In contrast, a 4.3% drop in DM intake for cows fed high- compared with low-starch diets based on grass silage (212 vs. 116 g starch/kg DM) has been reported [41]. Starch effect on feed intake can be mediated by a number of factors including starch fermentation rate, forage content of diet, amount of metabolic fuel absorbed from the rumen (for instance VFA), rumen pH, and rumen fermentation parameters [42,43]. It should be noted that the source of starch in diet and processing method can also contribute to the mixed results of different experiments. Hatew et al. [41] attributed the lower DM intake in high starch diets to increased propionate concentration in the rumen since hepatic oxidation of propionate influences DM intake [44]. Therefore, as in the aforementioned studies, no significant change was found in ruminal propionate concentration which is consistent with unaffected DM intakes.

In our experiment, digestibility of DM, OM, starch, and GE was higher and CP digestibility tended to be higher in cows fed HS than those receiving LS diets. Beckman and Weiss [45] observed linear increase in DM and OM digestibility as a result of reduction in NDF:starch ratio and linked it to replacement of highly digestible carbohydrates (i.e., starch) with low-digestible carbohydrates (i.e., NDF). In the current experiment, lower NDF and pdNDF digestibility along with improved starch digestibility beside the lack of effect on rumen pH as a factor influencing fiber digestion, might reflect the competition between rumen microbial population to utilize more easily nonstructural carbohydrates when available. As shown by the results of the microbiological analysis, the abundance of *Ruminobacter* sp. or *Selenomonas* sp. increased in high starch diets while the abundance of amylolytic Prevotella or cellulolytic bacteria Ruminococcus and Fibrobacter in LS-fed cows was not significantly different from that in HS-fed cows. Another explanation might be that observed differences in fiber digestibility reflect changes in the composition of diets; in HS diet rolled barley and ground wheat were used instead of sugar beet pulp and barley feed which might have different fiber characteristics.

It should be noted, however, that in some experiments where starchy feeds were replaced by fibrous by-products, there has been a confounding effect of both starch:NDF ratio and forage proportion of the diet [45]. However, in our experiment, the forage proportion in both LS and HS diets were fixed to remove such a source of difference between diets.

### 4.2. Methane Production and Rumen Fermentation

The oil supplementation to both low- and high-starch diets resulted in lower CH_4_ production compared with non-supplemented diets (475 vs. 552 g/d, on average). This is in line with the findings of the experiments that used oilseeds [29,46,47], or fish oil [30] to supplement dairy cow diets. In the current experiment, enteric CH_4_ emission (g/d) reduced by on average 4.7% for each 10 g/kg DM unsaturated oil in the diet. Similarly, Martin et al. [4] indicated that CH_4_ production reduces by 3.8% as a result of every 10 g/kg increase in dietary lipid supplementation. The most of this response in our experiment was caused by lower feed intake as CH_4_ yield (g/kg OMI) was not affected by the oil supplementation (average numerical reduction of 1.7% in CH_4_ yield). Ramin and Huhtanen [48] in a meta-analysis showed that 1 g/kg of DM increase in dietary EE concentration decreased CH_4_ yield by 0.043 L/kg of DM. The equivalent value in our experiment was on average 0.045 g/kg of OM intake which is very close to the reported value. Apparently both biohydrogenation of unsaturated fatty acids and amount of unfermentable organic matter introduced by oil to the diet should have caused the lower CH_4_ yield. Oil potential in reducing CH_4_ production seems to be a function of factors such as source of oil, fatty acid composition and level of supplementation in the diet [2,4]. Several mechanisms are known to influence the impact of fats and oils in reducing CH_4_ production including reduced OM fermentation in the rumen, unfavorable effects of C12:0 and C14:0 on protozoa community, and inhibition of methanogens by 18-carbon unsaturated fatty acids [4,49,50] and the competition for using hydrogen for biohydrogenation of unsaturated fatty acids. Patra [51] reported that C12:0 and C18:3 fatty acids are stronger inhibitors of methanogenesis compared with other fatty acids. Furthermore, as far as reduction in CH_4_ production is concerned, MUFA and PUFA are more effective than SFA in the diet. In this experiment, oil supplementation to both low- and high-starch diets resulted in higher intakes of both MUFA and PUFA compared with non-supplemented diets.

Unaffected CH_4_ yield for cows fed oil-supplemented diets in our experiment is consistent with the lack of differences in rumen pH, acetate, propionate, and VFA concentration and acetate:propionate ratio, although acetate and propionate concentrations and acetate:propionate ratio decreased numerically. However, oil supplementation is expected to not only reduce CH_4_ production but also lower acetate:propionate ratio. The non-significant change in acetate:propionate ratio observed in our experiment can be attributed to the lower DM and concentrate intakes, and the forage portion of our experimental grass silage-based diet. Changes in rumen fermentation pattern due to oil supplementation with diets based on restrictively fermented grass silage (using the silage additive based on formic acid) may be resistant to lipid supplementation [7].

Even though the intake of GE increased as a result of including oil in LS and HS diets, there were no significant differences between experimental treatments in terms of CH_4_ production as a percentage of GE intake while CH_4_ intensity calculated as g/kg milk or ECM decreased and tended to decrease by oil supplementation, respectively. In fact, the reduction in CH_4_ intensity indicates that net energy is partitioned more towards milk production, leading to lower CH_4_ intensity [3]. The values of CH_4_ production as a percentage of GE intake for un-supplemented diets (LS and HS; 7.25 and 7.04%) having DM intake of 23.0 and 23.5 kg/d are higher than 6.4% measured from dairy cows receiving rather similar diets with similar DM intake in respiration chambers (Bayat et al., unpublished data).

Our findings showed that starch level did not influence ruminal CH_4_ and CO_2_ production (g/d) and emission intensity (g/kg milk yield). Previous studies [39,40,41] have reported lower or tendencies towards lower daily CH_4_ production and CH_4_ emission intensity with the exception of Hatew et al. [41] reporting a non-significant CH_4_ emission intensity due to increasing dietary starch level. Increased starch content influenced rumen fermentation parameters, with a significant decrease in molar proportion of acetate and a tendency to reduce acetate:propionate ratio. Although high starch diets are expected to reduce acetate and increase propionate molar proportions and to lower CH_4_ production [4], this mechanism may not invariably apply to all experimental conditions. We did not observe any differences in rumen pH between cows fed low- and high-starch diets while it has been shown that high starch content lowers rumen pH, thereby limiting growth or activity of methanogens and cellulolytic bacteria [4]. The higher rumen ammonia N concentration due to higher starch level can be attributed to slightly higher dietary CP level whereas the higher rumen ammonia N concentration with oil-containing diets might have arisen from adding urea, which is highly rumen-degradable, to the diets in an attempt to make them isonitrogenous.

### 4.3. Rumen Microbial Ecology

Starch level had stronger effect on bacterial richness compared with those caused by inclusion of oil in the diet. Despite reports of toxic effects of oils on microbial community and, therefore, greater expected changes in the rumen microbial composition [52], this study did not observe significant oil effect on the reduction of alpha diversity in bacterial, archaeal or ciliate protozoan communities. Effects of oil on the microorganisms may depend on type and amount of oil, type of fatty acids in diet, and type of dietary forage fed. Bayat et al. [7] showed that supplementation of dairy cow diets with plant oils like rapeseed, safflower, linseed or myristic acid reduced CH_4_ production, with each type of oil having a different impact on bacterial community. Furthermore, Martin et al. [37] reported that adding extruded linseed to hay-based diets or corn silage-based diets for dairy cows reduced CH_4_ production without a notable change in abundance of rumen methanogens and cellulolytic bacteria. Looking at individual microbial taxa, addition of oil to both low- and high starch diets in this study, provided a suitable ecological niche for lipid hydrolyzing *Anaerovibrio* sp. [53] and Spirochaetes order PL-11B10. PL-11B10 was detected at significantly higher abundance in the diet with myristic acid supplement [7] and outside ruminants has been found positively correlated with methane production in methanogenic oil wells [54]. Nevertheless, our understanding of PL-11B10 ecology in rumen is limited. Oil supplementation also increased relative abundance of ciliate protozoa *Eudiplodinium-Eremoplastron* (AB536716). Similarly, a significant increase in *Eremoplastron dilobum* abundance was detected in an in vitro experiment with linseed oil but not rapeseed oil additive [55], suggesting that a positive or negative oil effect is depends on protozoa species and oil type.

High starch diets increased abundances of known starch utilizers like *Ruminobacter* sp., *Selenomonas* sp. or ciliate protozoa *Isotricha* sp. and *Entodinium* sp. that are involved in the utilization of non-structural polysaccharides and soluble sugars [53]. The Bacteroidetes family S24-7 was also enriched in HS diets. Current research suggests that members of S24-7 family are versatile with respect to complex carbohydrate degradation, but starch utilization trait is common to all family members and increased abundance of S24-7 is correlated with increased propionate production [56]. Inclusion of oil in LS and HS diets did not change total VFA, acetate, propionate, and butyrate but reduced CH_4_ production, suggesting lower H_2_ availability in these ruminal ecosystems. S24-7 was significantly more abundant in HSO diet, was strongly negatively correlated with acetate and tended (*p* = 0.054) to be positively (R = 0.49) correlated with propionate concentration in the rumen (data not shown). In addition to S24-7, also *Prevotella* sp., *Moryella* sp. and members from *Paraprevotellaceae* family were positively correlated with propionate concentration in the rumen (Figure 2). Given that propionate concentration was numerically the highest in HSO diet, these bacteria may have contributed to the sink in reduction of H_2_ availability for methanogenesis.

Low starch diets were enriched with bacteria directly or indirectly involved in fiber degradation. *Clostridium*, *Treponema*, (TM7) F16, (Tenericutes) RF39, and *Desulfovibrio* were found to be tightly attached to switchgrass [57] or wheat straw [58] during degradation process. In co-cultures with *Fibrobacter succinogenes*, *Treponema bryantii* has been shown to utilize soluble sugars released from cellulose degradation [53]. In addition to bacteria, ciliate protozoa *Ostracodinium* sp. and *Dasytricha* sp., are known to contain cellulolytic and hemicellulolytic activities, respectively, and were significantly enriched in LS diets. *Charonina ventriculi* is a holotrich protozoa not frequently observed in the rumen and with limited information about its metabolism. In our experiment, *Charonina ventriculi* was significantly enriched in LS diets and was negatively correlated with ammonia-N and butyrate, but positively correlated with acetate concentration in the rumen. Correlation profile of *Charonina ventriculi* was similar to *Treponema* sp. and *Clostridium* sp. (Figure 2) suggesting that these microorganisms require similar rumen conditions for thriving or are involved in similar metabolic processes.

*Methanobrevibacter gottschalkii* and *Mbb. ruminantium* were the predominant archaea without being significantly affected by starch level or oil supplement, although oil numerically reduced their abundance. *Methanobrevibacter* are hydrogenotrophic methanogens that convert H_2_ and CO_2_ produced by protozoa, bacteria, and fungi to CH_4_. In our study, numerical decreases in both *Mbb. gottschalkii* and *Mbb. ruminantium* correlated with decrease in CH_4_ intensity (g/kg milk) in oil supplemented diets. With higher abundances of bacteria *Moryella* sp., *Anaerovibrio* sp., (Bacteroidetes) S24-7, (Spirochaetes) PL11B10, *Selenomonas* sp., *Ruminobacter* sp., (Paraprevotellaceae) YRC22 and ciliate protozoa *Anoplodinium-Diplodinium* which had a tendency to be positively correlated with propionate concentration in the rumen, we can hypothesize having an ecosystem with less hydrogen available for methanogenesis.

Reduced daily CH_4_ emissions (g/d) were positively associated with reduction in *Entodinium caudatum.* Although *Entodinium* was the most abundant ciliate protozoa in all diets and smaller *Entodinium* spp. have been suggested to contribute more to CH_4_ production compared with larger protozoa in in vitro studies [59], a deeper subdivision of *Entodinium* into OTUs suggests functional versatility and differences in host dependency inside this genus Contrary to our results, Belanche et al. [60] investigated holotrich protozoa role in CH_4_ production compared to the natural flora and concluded that holotrichs were responsible for increased methanogenesis more than the entodiniomorphids. In our study, OTU affiliated with *Isotricha prostoma* was negatively correlated with daily CH_4_ emissions (OTU detected in 10 samples and therefore not included in Figure 2), while other OTUs affiliated with holotrichs did not produce significant associations with CH_4_ production. Discrepancies in observations could relate to the differences in the basal diet and dietary treatment as well as host impact on the general microbial community composition.

Based on the results of this experiment, it can be argued that inclusion of the mixture of fish oil and sunflower oil at 30 g/kg of diet DM does not have profound toxic effects on bacteria, archaea, or ciliate protozoa, and it is possible that the minor reduction in CH_4_ yield caused by inclusion of oil in diet may be linked more to the functional networks of microbiome possibly leading to a lower availability of hydrogen in the rumen which is required for CH_4_ production.

## 5. Conclusions

Overall, the results of this experiment show that starch level modified rumen fermentation and nutrient digestibility without influencing DM intake or methane emissions. Inclusion of unsaturated oil mixture (sunflower and fish oils, 2:1 *w*/*w*) reduced DM intake and some ruminal methane emission indices without influencing rumen fermentation characteristics or nutrient digestibility. The findings of this experiment show that feeding more starch originating from concentrate portion instead of fiber at a moderate level in dairy cow diets does not favor lower methane production, and oil supplementation is similarly effective on reducing methane in low- and high-starch diets. Therefore, our hypothesis that starch level and oil supplementation would have synergistic effects on CH_4_ emission could not be proved as increasing dietary starch level did not influence CH_4_ emission whereas oil supplementation did. Inclusion of moderate amount of the unsaturated oil mixture in the diet did not have profound toxic effects on bacteria, archaea, or ciliate protozoa, which is in line with the minor effect on methane yield.

## Figures and Tables

**Figure 1 animals-11-01310-f001:**
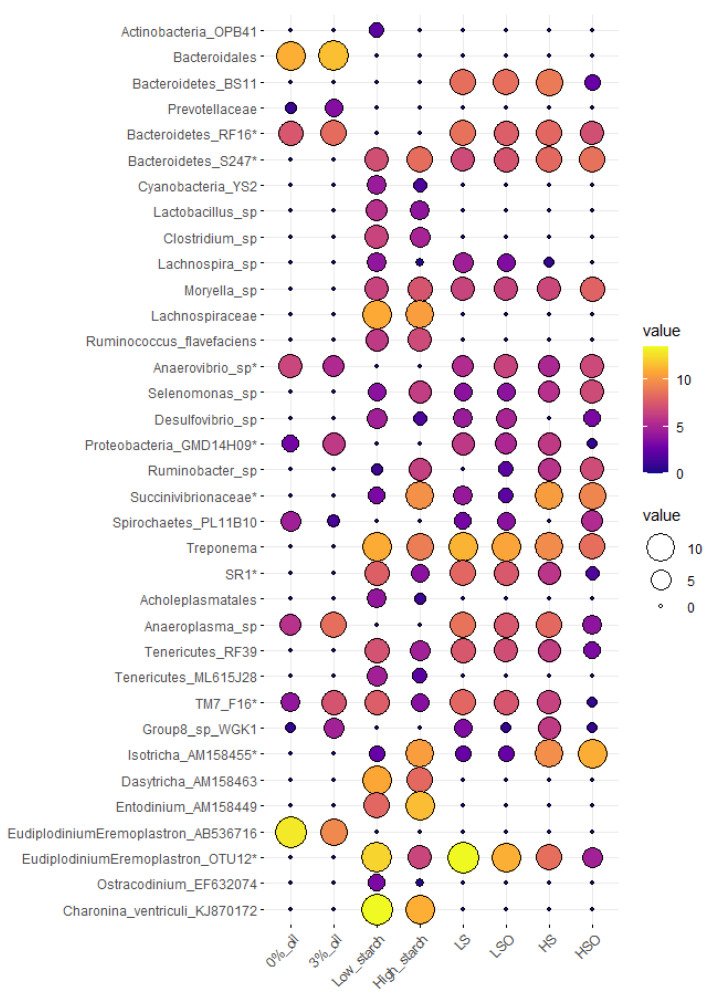
Mean abundance plot of microbial taxa that were significantly (*p* < 0.05) affected by oil, starch, or were significantly different in dietary pairwise comparisons. Microbial taxa tested are represented on *Y*-axis, while treatment groups tested are presented on the *X*-axis. Microbial values are presented as CSS normalized and log-transformed abundance data. An increase in bubble size and transition from purple to yellow color corresponds with the increase in abundance of a particular OTU. Microbial taxa with (*) are significant after FDR correction (FDR < 0.05).

**Figure 2 animals-11-01310-f002:**
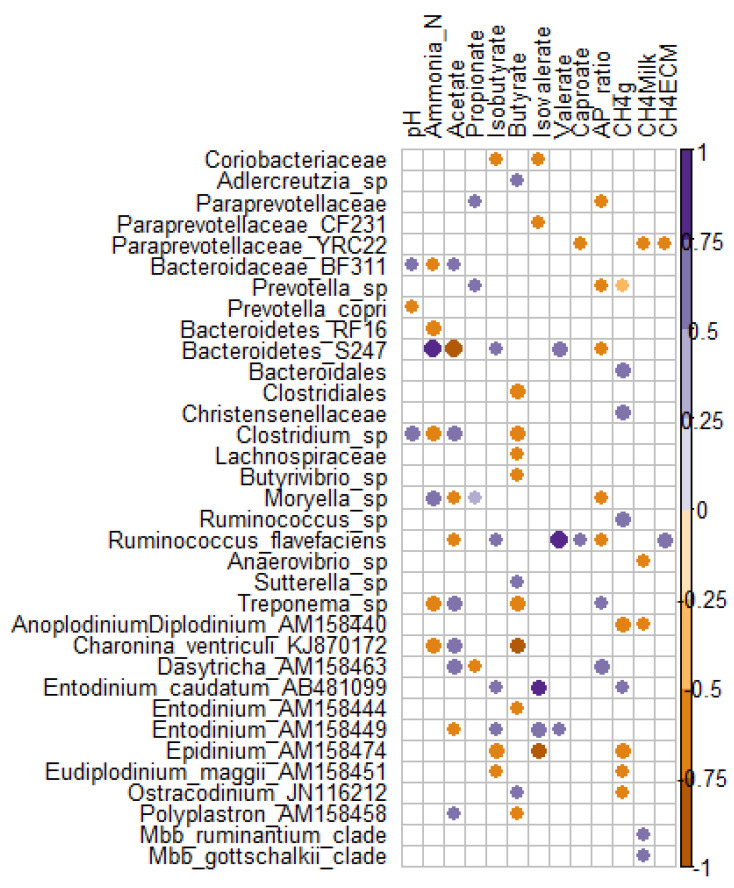
Spearman correlation plot between microbial taxa and rumen fermentation parameters. Only significant (*p* < 0.05) positive (purple) and negative (orange) correlations are presented.

**Table 1 animals-11-01310-t001:** Formulation and chemical composition of experimental diets.

	Diet ^1^			
	LS	LSO	HS	HSO
Inclusion rate (g/kg DM)				
Grass silage	550	550	550	550
Rolled barley	0	0	85	77
Ground wheat	0	0	255	231.5
Sugar beet pulp	255	231.5	0	0
Barley feed	85	77	0	0
Rapeseed meal, solvent-extracted	80	80	80	80
Urea	0	1.5	0	1.5
Oil mixture ^2^	0	30	0	30
Vitamin and mineral Pre-mix ^3^	30	30	30	30
Chemical composition (g/kg DM unless stated)				
DM (g/kg as fed)	536	539	528	532
OM	926	928	941	942
CP	159	160	165	165
FA	23.5	51.4	22.9	50.8
NDF	430	417	344	339
pdNDF	348	337	276	271
WSC	44.8	42.3	27.5	26.6
Starch	16.1	15.0	202	184
GE (MJ/kg DM)	18.3	18.9	18.4	19.0

^1^ Refers to diets (forage to concentrate ratio 55:45, on a DM basis) consisting of low or high levels of starch supplemented with 0 (LS and HS, respectively) or 30 g of unsaturated fatty acid mixture/kg diet DM (LSO and HSO, respectively). ^2^ Mixture of sunflower oil and fish oil (2:1 *w*/*w*). ^3^ Onni-Kivennäinen, Melica Finland Ltd., Vaasa, Finland) declared as containing: (g/kg) calcium, 205; magnesium, 72; sodium, 85; phosphorus, 27; zinc, 1.46; manganese, 0.35; copper, 0.27; (mg/kg) iodine, 39; cobalt, 27; selenium, 20; (IU/g) retinyl acetate, 120; cholecalciferol, 25; and dl-α tocopheryl acetate, 0.34.

**Table 2 animals-11-01310-t002:** Effect of dietary starch level and a mixture of unsaturated fatty acids on nutrient intake in lactating cows.

Intake (kg/d Unless Stated)	Treatment ^1^				SEM	*p*-Value ^2^		
	LS	LSO	HS	HSO		S	O	S × O
Silage DM	13.1	11.9	13.4	11.9	0.26	0.36	<0.001	0.56
Concentrate DM	9.9	8.5	10.1	8.5	0.19	0.57	<0.001	0.35
Oil	-	0.60	-	0.61	0.006	-	<0.001	-
DM	23.0	21.0	23.5	21.0	0.43	0.43	<0.001	0.45
OM	20.7	18.9	21.5	19.2	0.39	0.087	<0.001	0.38
CP	3.66	3.35	3.86	3.44	0.062	0.013	<0.001	0.27
NDF	10.0	8.80	8.22	7.23	0.174	<0.001	<0.001	0.51
pdNDF	8.06	7.11	6.58	5.78	0.140	<0.001	<0.001	0.49
WSC	1.01	0.87	0.64	0.55	0.027	<0.001	0.007	0.43
Starch	0.36	0.31	4.57	3.72	0.036	<0.001	<0.001	<0.001
GE intake (MJ/d)	421	396	434	400	7.8	0.18	0.002	0.42
FA intake (g/d)								
12:0	0.44	0.74	0.46	0.75	0.012	0.12	<0.001	0.63
14:0	2.1	16.1	2.0	16.1	0.21	0.93	<0.001	0.89
16:0	88	131	86	128	2.2	0.067	<0.001	0.81
*cis*-9 16:1	3.2	18.1	3.3	18.2	0.24	0.78	<0.001	0.95
18:0	7.0	27.3	7.3	27.5	0.37	0.42	<0.001	0.95
*cis*-9 18:1	74	188	73	186	2.7	0.38	<0.001	0.95
*cis*-11 18:1	13.1	20.0	13.0	19.8	0.34	0.47	<0.001	0.75
*cis*-9, *cis*-12 18:2	167	373	171	375	5.4	0.41	<0.001	0.73
20:0	5.85	6.83	5.85	6.77	0.136	0.70	<0.001	0.69
18:3n–3 ^3^	145	136	145	134	3.1	0.65	0.003	0.64
22:0	4.31	6.90	4.07	6.67	0.114	0.015	<0.001	0.99
*cis*-11 22:1	0.02	2.17	0.01	2.17	0.028	0.91	<0.001	0.82
20:5n–3	0.0	31.5	0.0	31.6	0.41	0.85	<0.001	0.85
24:0	3.61	4.18	3.00	3.63	0.076	<0.001	<0.001	0.55
22:5n–3	0.00	3.63	0.00	3.64	0.048	0.85	<0.001	0.85
26:0	2.92	2.76	3.04	2.81	0.064	0.072	0.002	0.48
22:6n–3	0.0	20.2	0.0	20.3	0.26	0.85	<0.001	0.85
Unidentified	0.92	2.90	1.09	3.04	0.041	0.002	<0.001	0.59
Σ 25–30	11.7	10.7	11.9	10.8	0.25	0.37	0.001	0.56
SFA	127	211	124	207	3.4	0.10	<0.001	0.85
MUFA	118	267	113	263	3.9	0.11	<0.001	0.98
PUFA	294	568	300	570	8.8	0.43	<0.001	0.72
Total FA	539	1049	537	1042	16.0	0.69	<0.001	0.81

^1^ Refers to diets (forage to concentrate ratio 55:45, on a DM basis) consisting of low or high levels of starch supplemented with 0 (LS and HS, respectively) or 30 g of unsaturated fatty acid mixture/kg diet DM (LSO and HSO, respectively). Values are LS means and pooled SEM for *n* = 4. ^2^ S = effect of starch in the diet; O = effect of oil mixture; S × O = effect of interaction between S and O. ^3^ Co-elutes with cis-11 20:1.

**Table 3 animals-11-01310-t003:** Effect of dietary starch level and a mixture of unsaturated fatty acids on milk yield and composition in lactating cows.

	Treatment ^1^				SEM	*p*-Value ^2^		
	LS	LSO	HS	HSO		S	O	S × O
Yield								
Milk (kg/d)	30.9	31.3	31.1	30.4	1.55	0.73	0.85	0.55
ECM (kg/d)	30.2	30.0	31.3	27.8	1.87	0.58	0.11	0.14
Fat (g/d)	1242	1230	1266	1093	85.2	0.26	0.087	0.12
Protein (g/d)	1021	981	1079	924	51.0	0.99	0.030	0.15
Lactose (g/d)	1322	1374	1383	1358	101.3	0.71	0.82	0.52
Milk composition (g/kg)								
Fat	40.1	39.4	40.5	35.8	1.16	0.13	0.024	0.072
Protein	33.1	31.5	34.7	30.6	0.94	0.68	0.008	0.15
Lactose	42.7	43.9	44.3	44.3	1.45	0.35	0.54	0.52
ECM/DMI	1.31	1.43	1.33	1.32	0.075	0.42	0.31	0.28

^1^ Refers to diets (forage to concentrate ratio 55:45, on a DM basis) consisting of low or high levels of starch supplemented with 0 (LS and HS, respectively) or 30 g of unsaturated fatty acid mixture/kg diet DM (LSO and HSO, respectively). Values are LS means and pooled SEM for *n* = 4. ^2^ S = effect of starch in the diet; O = effect of oil mixture; S × O = effect of interaction between S and O.

**Table 4 animals-11-01310-t004:** Effect of dietary starch level and a mixture of unsaturated fatty acids on nutrient digestibility in lactating cows.

	Treatment ^1^				SEM	*p*-Value ^2^		
	LS	LSO	HS	HSO		S	O	S × O
Digestibility (g/kg or otherwise stated)								
DM	691	699	712	720	3.1	0.001	0.13	0.98
OM	706	714	726	733	3.4	0.002	0.15	0.96
CP	674	682	682	719	7.3	0.063	0.058	0.20
NDF	608	615	562	563	5.9	<0.001	0.61	0.76
pdNDF	751	761	701	704	7.2	<0.001	0.55	0.76
Starch	858	805	927	928	14.3	0.002	0.20	0.18
GE (kJ/MJ)	680	692	699	715	3.4	0.003	0.019	0.66

^1^ Refers to diets (forage to concentrate ratio 55:45, on a DM basis) consisting of low or high levels of starch supplemented with 0 (LS and HS, respectively) or 30 g of unsaturated fatty acid mixture/kg diet DM (LSO and HSO, respectively). Values are LS means and pooled SEM for *n* = 4. ^2^ S = effect of starch in the diet; O = effect of oil mixture; S × O = effect of interaction between S and O.

**Table 5 animals-11-01310-t005:** Effect of dietary starch level and a mixture of unsaturated fatty acids on rumen fermentation characteristics in lactating cows.

	Treatment ^1^				SEM	*p*-Value ^2^		
	LS	LSO	HS	HSO		S	O	S × O
pH	6.79	6.68	6.64	6.67	0.061	0.24	0.56	0.32
Ammonia N (mmol/L)	5.60	6.16	7.67	9.17	0.306	<0.001	<0.001	0.027
Total VFA (mmol/L)	100	100	110	102	2.6	0.056	0.21	0.18
Molar proportions (mmol/mol)								
Acetate	681	681	653	637	5.8	<0.001	0.23	0.20
Propionate	179	181	186	203	9.8	0.18	0.37	0.50
Butyrate	99.3	101	111	112	4.41	0.016	0.77	0.96
Isobutyrate	7.77	6.97	9.41	9.09	0.695	0.014	0.35	0.67
Valerate	15.3	14.1	17.5	17.5	0.47	<0.001	0.28	0.21
Isovalerate	11.4	9.81	15.2	14.4	1.76	0.004	0.26	0.68
Caproate	6.93	6.35	8.25	7.85	0.544	0.034	0.38	0.87
Molar ratio								
Acetate:Propionate	3.81	3.77	3.56	3.17	0.193	0.070	0.31	0.41

^1^ Refers to diets (forage to concentrate ratio 55:45, on a DM basis) consisting of low or high levels of starch supplemented with 0 (LS and HS, respectively) or 30 g of unsaturated fatty acid mixture/kg diet DM (LSO and HSO, respectively). Values are LS means and pooled SEM for *n* = 4. ^2^ S = effect of starch in the diet; O = effect of oil mixture; S × O = effect of interaction between S and O.

**Table 6 animals-11-01310-t006:** Effect of dietary starch level and a mixture of unsaturated fatty acids on ruminal methane and carbon dioxide emissions in lactating cows.

	Treatment ^1^				SEM	*p*-Value ^2^		
	LS	LSO	HS	HSO		S	O	S × O
Enteric CH_4_								
g/d	551	471	553	478	39.4	0.89	0.051	0.93
g/kg OMI	26.8	25.0	25.9	25.0	1.95	0.79	0.46	0.80
g/kg OMD	38.2	35.1	36.9	33.1	2.53	0.46	0.15	0.87
g/kg milk	17.8	15.1	17.8	15.9	0.74	0.64	0.015	0.58
g/kg ECM	18.2	15.7	17.8	17.4	0.70	0.37	0.067	0.16
% of GEI	7.25	6.56	7.04	6.59	0.521	0.86	0.26	0.80
Enteric CO_2_								
g/d	6109	4488	4937	4846	544.8	0.48	0.16	0.21
g/kg OMD	424	335	330	336	34.7	0.23	0.27	0.22
g/kg milk	198	144	159	158	12.8	0.38	0.074	0.086
g/kg ECM	202	149	160	173	12.2	0.46	0.15	0.036

^1^ Refers to diets (forage to concentrate ratio 55:45, on a DM basis) consisting of low or high levels of starch supplemented with 0 (LS and HS, respectively) or 30 g of unsaturated fatty acid mixture/kg diet DM (LSO and HSO, respectively). Values are LS means and pooled SEM for *n* = 4. ^2^ S = effect of starch in the diet; O = effect of oil mixture; S × O = effect of interaction between S and O.

**Table 7 animals-11-01310-t007:** Alpha diversity estimates for bacteria, archaea, and ciliate protozoa.

	Diversity	Treatment ^1^				SEM	*p*-Value ^2^		
	Estimate	LS	LSO	HS	HSO		S	O	S × O
Bacteria	Shannon	2.21	2.22	2.18	2.02	0.104	0.23	0.45	0.38
	Simpson	0.728	0.732	0.738	0.662	0.035	0.29	0.21	0.17
	Richness	68.5	68.4	63.3	62.3	2.019	0.014	0.73	0.78
	Evenness	0.519	0.524	0.522	0.486	0.022	0.36	0.42	0.30
Archaea	Shannon	1.17	1.18	0.96	1.01	0.135	0.13	0.79	0.86
	Simpson	0.594	0.633	0.515	0.536	0.059	0.15	0.60	0.87
	Richness	6.81	6.33	7.69	5.61	0.723	0.89	0.064	0.21
	Evenness	0.581	0.589	0.437	0.545	0.056	0.10	0.28	0.34
Ciliate protozoa	Shannon	2.15	2.26	2.23	2.12	0.088	0.80	0.85	0.22
	Simpson	0.840	0.854	0.855	0.836	0.009	0.89	0.74	0.071
	Richness	25.9	27.7	26.0	27.5	1.626	0.98	0.31	0.93
	Evenness	0.628	0.648	0.663	0.613	0.014	0.97	0.24	0.022

^1^ Refers to diets (forage to concentrate ratio 55:45, on a DM basis) consisting of low or high levels of starch supplemented with 0 (LS and HS, respectively) or 30 g of unsaturated fatty acid mixture/kg diet DM (LSO and HSO, respectively). Values are LS means and pooled SEM for *n* = 4. ^2^ S = effect of starch in the diet; O = effect of oil mixture; S × O = effect of interaction between S and O.

## Data Availability

The data presented in this study will be made available on reasonable request from the corresponding author. Microbial sequencing data is submitted to Dryad Digital Repository with doi:10.5061/dryad.pvmcvdnk7.

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
