# Peer review of "Effects of Starch Level and a Mixture of Sunflower and Fish Oils on Nutrient Intake and Digestibility, Rumen Fermentation, and Ruminal Methane Emissions in Dairy Cows"

_animals, 2021, doi:10.3390/ani11051310_

Round 1
Reviewer 1 Report
The trial aimed to assess the effect of two different levels of starch, with or without an unsaturated oil mixture, on the performances of lactating dairy cows. This trial is not exclusively focused on the production of methane; more effects were investigated, therefore I suggest a more appropriate title.
Line 85 - Invert HS and LS starch amount
Line 86 and Table 1 - explain in the text what LSO and HSO mean
Table 1 - FA is Fatty Acid?; if so, I suggest to add the ether extract value (EE) of diets. The sources of fat in the diet (EE, supplementary fat or both) are important in prediction models of CH4 production
Line 108 - TMR dimension influence the passage rate of feed particles, thereby reducing diet digestibility and amount of fermented substrate per unit of intake. A high passage rate may cause a decline of methane emission. Did you measure the particle size of TMR? Did both diets have the same particle size?
Lines 187-191 - Explain calculations of intensity of methane and carbon dioxide emissions as shown in Table 5 (g/kg OMI, g/kg OMD, g/kg milk, g/kg ECM, % CEI), and their meaning
Table 3 - split into 2 tables (effect of diets on milk and effects of diets on digestibility)
Reviewer 2 Report
This paper describes the effect of starch and oil (a fish oil : canola oil mixture) on the methane emission in dairy cows. The experiment in itself seems a well setup experiment that merits publication. However the current paper would benefit from several improvements, which are suggested below. It is advised to thoroughly revise the first section (4.1) of the discussion according to suggestions below.
L1-4 Why does the title say there is no effect of oil supplementation on ruminal methane emission, while the results say differently?
L27. Oil decreased CH4 per kg milk significantly according to table 5. The tendency is for CH4 per kg ECM.
L36, I would not mention the increase in CP digestibility here, as I believe it is caused by ingredient selection. (especially for oil, where urea is added).
L83. It states isonitrogenous diets. However Table 1 gives a slightly higher CP level for the High Starch diets. Normally I would not have bothered with a 5 g/kg DM differences as it is only a 3% difference. However Rumen ammonia is higher with both starch and Oil supplementation, both are unexpected. I believe the higher rumen ammonia by high starch is probably explained by the higher CP level of these diets, where the higher rumen ammonia by oil, would be explained by the CP exchange of real protein versus highly degradable urea. This needs to be addressed somewhere in the discussion.
I would say something like intended to be isonitrogenous, because I do not think they actually were. This of course can happen in this type of research, but it should be addressed properly.
L101: The oil replaced sugar beet pulp and barley feed, with the reduction in protein balanced by additional urea. Interesting choice urea, why not put some more rapeseed meal? Although in that case shifts in ingredients would be larger as also a larger proportion of barley feed and sugar beet pulp would need to be switched. Maybe spend a few words on the rationale behind the ingredient exchanges made.
L102-103: Please make a reference to a paper for these experiences, then we can try and follow this reference and understand the rationale. Alternatively describe those experiences shortly.
L122-123: iNDF determination is interesting, but the potential NDF degradability (pdNDF) calculated by using iNDF is not reported in table 1, whereas it is used in table 2.
Line 146. to separate urine from feces.
Line 148 same analysis? also iNDF? Because that would be interesting, then also digestibility could be determined by marker method (iNDF) and total collection (or would that be short communication).
Lines 153-166. From the end, I understand SF6 is used as a marker gas. Would be good to mention that in the beginning of this section and use one or two lines on describing that procedure briefly as well.
L193-201: I would expect that if the interaction term was significant the different treatments would have been indicated with superscripts in the tables.
L211-216. The individual taxa data seem to be analyzed in the same way as the other data, as described in L193-201. But that is not completely clear. But for sure they are reported differently, see comments in table 6
Table 1
Include pdNDF
Table 2
There is a superscript 3 at the DM... but under the table the 3 refers to another superscript saying it co-eludes with cis-11 20:11. So the superscript at DM must refer to something else.
Concentrates as fed? Or also DM?
Actually if diet is fed as a TMR, why not give total DM intake? Should statistics not be done on total TMR?
In lines 115-116 it is described that daily feed intake was calculated by subtracting refusals by daily offered feeds. But it was not mentioned whether refusals were considered to be exactly the same as TMR, or maybe wetter because of saliva, or maybe different because of feed selection by cows. I can only imagine the sense of reporting and statistically analyzing the separate silage and concentrate intakes if there are some assumptions on feed selection of cows. Otherwise only the TMR DM intake should be analyzed. Of course the separate silage and concentrate intakes can still be reported, but the analysis should be on total TMR.
Lines 268-270. Differences in ammonia between treatments is mentioned here, but not discussed in the results. I have given some comments above (at L83) regarding why these differences may be in the direction as they are. If the ammonia differences are mentioned explicitly in the results it is good to also mention these in the discussion.
Table 6
It is surprising that the method of presentation has changed for this table. In other tables just the 4 treatments were presented, now the oil and starch effect are presented separately as well. I doubt all these presentations are necessary, also because there is no interaction term. If this presentation form is kept, then please switch the columns of yes and no within both the oil and starch, as now they are in the different order than the treatments are ordered.
It may be an effect of the statistical analysis, but the interaction term has disappeared and a diet effect has entered. If the different type of data allows (I am not a statistical, so I cannot judge) it would be good if the interaction term could return.
The fonds for the different numbers seem not the same, this is really weird reading.
L351. It is stated that: “Diet had significant effect on 17 bacterial, 1 archaeal and 2 ciliate protozoan taxa (Figure 1).”. This is confusing, there are 35 taxa mentioned in Figure 1. Which according to the caption of figure one are the significant ones. However, 17 + 1 +2 is 20. The discrepancy between 20 and 35 is not clear.
L362-366; The difference in variance components for the various taxa is interesting. However as this section is described now it is stated that Metadinium sp is significantly more abundant in cow 1 and etc. However from the supplemental data we only see the variance components, without statistics or individual cows. So this section is hard to verify using the table in the supplementary data. Please reformulate or add statistics on the supplementary table.
Figure 1.
I do not understand why these data cannot be presented as the previous tables. If I understand correctly, according to the statistics ANOVA was done on these data, just like the other data (except for the diversity estimates). So why is this figure presented like it is, and not like (maybe a bit more boring) table with numerical values, with statistics on the effects of Starch level, oil level, and their interaction.
Figure 2
In this table using the colors is much more insightful than using numbers, nice!
L409-412. I agree that oils might inhibit digestibility of OM and NDF in the rumen, however this was not shown in this paper itself. So suggesting this needs to be connected to the lack of effect on digestibility in this paper.
L420. I went and checked this paper ((30) Vafa, T.S.; Naserian, A.A.; Moussavi, A.R.H.; Valizadeh, R.; Mesgaran, M.D. Effect of supplementation of fish and canola oil in 733 the diet on milk fatty acid composition in early lactating holstein cows. Asian-australas. J. Anim. Sci. 2012, 25, 311.), as I did not understand whether the 20 and 36.7 g EE/kg DM were added or the final levels. But the fat levels as are reported in the paper are: 27.8; 46.2; 46.7, and 45.3 g EE/kg DM for Control, Fish oil, :Fish oil & Canola oil, and Canola oil respectively, leading to differences in EE between treatments of below 20 g EE /kg DM. So I cannot trace these values mentioned in the current paper. Additionally in paper 30 the DMI is reported significantly lower for Fish oil (but indeed not for Canola oil).
L421-423. As much as I can agree with this statement, in essence this statement does not really help with explaining the current results. So what are the effect on DMI of oil content, diet composition, source of oil and type of basal diet, and how have they influences the current results?
L425. It would be good to include a reference to these previous experiences. I agree milk yield was not changed. ECM was not changed, but actually, it is just the milk fat content that changes. Maybe mention that, rather than the discussion on the ECM which is a calculated value derived from milk production and milk composition. You were successful in maintaining milk yield while depressing milk fat concentration! Focus on that, not on the lack of response in ECM.
L431-432. It is surprising that the first reference included here reports an increase in digestibility, while in lines 409-412 a decrease in digestibility was used as an argument for the decreased DMI. That is contradictory and should be addressed.
L438-440. This again (like in line 421-423) is maybe correct, but it does not really help in interpretation of the current results as the direction of the effects and how they interact are not reported, just that they have “an” effect.
L443-447. The sentence: “The unchanged DM intake due to higher starch level in this experiment, which is most probably due to insufficient differences in starch intakes between the treatments, is consistent with the findings of a meta-analysis of 414 treatment means revealing that starch content had no effect on DM intake [41] as well as Pirondini et al. [38] and Philippeau et al. [42].”, contains 2 contradictions. Contradiction nr 1: unchanged DMI due to insufficient differences in starch intake... seems odd, this is a bit chicken or the egg. But actually there is quite a big difference in starch intake, of 3.5-4.2 kg/day that is significant... so this is a bit strange. Contradiction nr 2: First it is stated that the lack of DMI is because starch intake was insufficiently different... this assumes that if the difference in starch intake would have been larger, one would have seen a difference in DMI. However this is then disproven by the reference to the meta-analysis, where no effect of starch on DMI was found.
L458: No! NDF and pNDF digestibility in Table 3 is Lower, not higher for HS.
L462-464; How does a replacement of soybean hulls by corn meal hulls reduce DM and OM digestibility (CP and NDF I can understand)? Does not sound logical, one would expect the starch in corn meal to be more digestible still than the NDF in soybean hulls right?
L472-473. My money is on the ingredient effect. The exchange is mainly a high pectin low ADL type of root fiber by a low pectin cereal fiber with higher ADL. Not surprised that NDF degradability is lower for the HS diets.
L484: CH4 yield in the text is referenced as g/kg DM intake. However in the table it is g/kg OMI... why not also include g/kg DMI. Also the difference between DMI and OMI is only Ash, which does not result in CH4 production. Thus why would g/kg DMI be different and g/kg OMI not (line 488).
L488: Why an especially lower concentrate intake? The diets were fed as TMR. The decrease in DMI should be equal for silage and concentrate. I did some simple calculations and numerically indeed the percentage of concentrate seems to be lower (but not especially lower) for LSO and HSO. This however should not be the case. This would mean there would be selection against concentrates with oil addition? Or was the oil itself not regarded as concentrate in the calculations?
Table 1: Recalculation of the concentrate percentage.
|
LS |
LSO |
HS |
HSO |
|
|
DM Silage |
13,1 |
11,9 |
13,4 |
11,9 |
|
concentrates |
9,9 |
8,5 |
10,2 |
8,5 |
|
total DMI |
23 |
20,4 |
23,6 |
20,4 |
|
% silage |
57,0 |
58,3 |
56,8 |
58,3 |
|
% concentrate |
43,0 |
41,7 |
43,2 |
41,7 |
Line 507.... now it says unaffected CH4 yield, while in line 484-485 it says that CH4 yield was reduced.... confusing!
L480 and further down: What a change in writing style. Please synchronize writing style, preferably towards the style as section 4.2 is written in. In section 4.2 the text reads smoother and the effects are discussed in a more integrated way.
L633-634: Therefore, the effect of increasing dietary starch level and oil supplementation on methane emissions was not additive. Why not? There was no significant interaction. Starch had no effect, but oil did... so still additive I would think: 0 for starch and x for oil... still 0+x = x in all cases.
References
Why the different font size in reference nr 10 and reference nr 61.
Round 2
Reviewer 2 Report
The paper has been much approved and the authors have in great detail processed the remarks made and/or answered the questions.
Re-reading my comments, I apologize for the questions on the statistics on the individual concentrate and forage intake. Indeed the DM intake and oil intake were in the same table.
I appreciate adding the table in the supplementary materials supporting the data in Figure 1. For me it gives more detail and I understand it easier, although I will admit it takes more size than the figure, so I agree with the authors that it is a matter of preference.
Author Response
Thank you